# Omnidirectional-Sensor-System-Based Texture Noise Correction in Large-Scale 3D Reconstruction

**DOI:** 10.3390/s24010078

**Published:** 2023-12-22

**Authors:** Wenya Xie, Xiaoping Hong

**Affiliations:** The School of System Design and Intelligent Manufacturing, Southern University of Science and Technology, Shenzhen 518055, China; hongxp@sustech.edu.cn

**Keywords:** imaging sensor, texture noise correction, frame fusion, voxel hashing, 3D reconstruction

## Abstract

The evolution of cameras and LiDAR has propelled the techniques and applications of three-dimensional (3D) reconstruction. However, due to inherent sensor limitations and environmental interference, the reconstruction process often entails significant texture noise, such as specular highlight, color inconsistency, and object occlusion. Traditional methodologies grapple to mitigate such noise, particularly in large-scale scenes, due to the voluminous data produced by imaging sensors. In response, this paper introduces an omnidirectional-sensor-system-based texture noise correction framework for large-scale scenes, which consists of three parts. Initially, we obtain a colored point cloud with luminance value through LiDAR points and RGB images organization. Next, we apply a voxel hashing algorithm during the geometry reconstruction to accelerate the computation speed and save the computer memory. Finally, we propose the key innovation of our paper, the frame-voting rendering and the neighbor-aided rendering mechanisms, which effectively eliminates the aforementioned texture noise. From the experimental results, the processing rate of one million points per second shows its real-time applicability, and the output figures of texture optimization exhibit a significant reduction in texture noise. These results indicate that our framework has advanced performance in correcting multiple texture noise in large-scale 3D reconstruction.

## 1. Introduction

In recent years, 3D imaging sensors have rapidly evolved towards higher resolutions, greater frame rates, and wider fields of view, thus propelling the growth and application of 3D reconstruction technologies, including digital tourism, virtual reality (VR), and augmented reality (AR). However, the process of 3D reconstruction often confronts numerous types of texture noise, a substantial part of which results from the inherent sensor limitations, such as luminance overflow of cameras attributed to dynamic range constraints. Especially for the highlight phenomenon caused by specular reflection, color information is completely lost, making it challenging to achieve full recovery through conventional image processing techniques, as shown in Figure 1a. Another type of texture noise is frame-to-frame color inconsistency caused by variations in the intensity of the light source or changes in the relative position of the light source to the camera, as shown in Figure 2. In addition to the texture noise caused by sensor limitations and illumination factors, occluding objects such as persons or animals that enter the field of view can also introduce texture noise. Besides multiple texture noise, the 3D reconstruction process also needs to handle massive amounts of data when reconstructing large-scale scenes.

Considerable research effort has focused on resolving texture noise, particularly in specular highlight removal. Techniques for specular highlight removal are mostly based on the dichromatic reflection model, which represents an image as a linear superposition of the specular reflection component and the diffuse reflection component. These methods include those found in [1,2,3,4,5,6]. Some approaches, like [7], even estimate light source positions for a more effective highlight removal. In addition to traditional techniques, the field is witnessing a significant rise in the development of learning-based methods for highlight noise removal, such as [8]. To tackle color inconsistencies, studies like [9,10,11] have explored local image translations and gradient domain techniques for seam smoothing. However, these solutions typically specialize in image processing, without fully integrating into 3D reconstruction workflows, and often cater to specific datasets, limiting their wider use. Given these limitations, there is a pressing need for an all-encompassing and efficient approach to eliminate various texture noise, crucial for generating high-quality textures in large-scale 3D reconstructions.

In this paper, we propose an omnidirectional-sensor-system-based framework for 3D reconstruction in large-scale scenes, with special emphasis on eliminating texture noise caused by sensor limitations and environmental disturbance. The omnidirectional sensor system, comprising a LiDAR unit and a camera with a 360-degree field of view, is selected for its ability to achieve extensive informational overlap across frames. This comprehensive coverage is critical for the advanced texture optimization process that follows. The process of the 3D reconstruction framework involves three stages: data organization, geometry reconstruction, and texture optimization, as shown in Figure 3.

The first stage is data organization, where we obtain a colored point cloud with luminance value through color space conversion, point cloud coloration, and multiframe registration. The color space conversion aims to obtain the luminance value of each RGB image pixel, which is essential for the frame fusion process during the texture optimization phase. Next, in point cloud coloration, we utilize predetermined intrinsic and extrinsic matrices from an existing calibration method to project the point cloud onto images, thereby retrieving and applying the corresponding color and luminance value to the point cloud. Finally, in multiframe registration, we transform all frames into a unified coordinate system using classical registration methods to obtain a dense colored point cloud annotated with its frame origin sequence number.

The second stage is geometry reconstruction. The main challenge for large-scale scene reconstruction is the computation of a substantial amount of input data. To address this issue, in the second stage, we implement an effective method proposed by Nießner et al. that accelerates computation speed and enhances memory efficiency [12]. The key to this method is the use of a hash table, which allows fast retrieval of data storage. The data are organized in a two-level voxel data structure, which not only improves retrieval efficiency but also ensures high resolution. This structure also offers advantages in the texture rendering stage, as it provides efficient data retrieval for color optimizations.

In the final stage, the objective is to obtain accurate texture from high-resolution RGB images in a universal method, mitigating texture noise in all the aforementioned cases: specular highlight, color inconsistency, and object occlusion. To achieve this, we propose frame-voting rendering and neighbor-aided rendering mechanisms for texture optimization. The frame-voting mechanism integrates frames from different viewpoints utilizing a ‘minority conforms to the majority’ rule at the voxel level, which removes color values significantly deviating from the overall luminance level and reduces color discrepancy between frames. The neighbor-aided mechanism is designed to address challenging situations where the points number is insufficient in a voxel for texture self-optimization, in which we borrow color information from neighboring voxels to enhance the texture.

In summary, our main contributions are outlined as follows:We propose a comprehensive 3D reconstruction framework based on an omnidirectional sensor system for large-scale scenes. The framework includes data organization, geometry reconstruction, and texture optimization.We propose a frame-voting rendering mechanism in texture noise correction by integrating multiple frames according to the luminance values, which eliminates texture noise such as specular highlight, frame color inconsistency, and object occlusion.We propose a neighbor-aided rendering mechanism to optimize color for certain voxels that has insufficient points for texture self-optimization, by using convincing color information from neighboring voxels.

## 2. Related Work

### 2.1. Imaging Sensors

The most common choice for 3D reconstruction sensors can be divided into 2D cameras, RGB-D cameras, and camera-LiDAR integrated systems. There are several methods to achieve 3D reconstruction utilizing 2D cameras. One of the methods is based on binocular disparity, which recovers the depth information by using two images captured from a slightly different position [13]. Another method is based on motion parallax, which perceives depth information based on the relative movement between the camera and the scene [14]. Additionally, structure from motion (SfM) is also a typical technique, detailed in [15], which uses a series of two-dimensional images of a scene to reconstruct its 3D structure.

With the advancement of the RGB-D camera, the acquisition of depth information became much easier. Lindner et al. [16] proposed a fast reconstruction approach with photonic mixing device (PMD) technology, a type of RGB-D camera based on time of flight (TOF). In 2010, Microsoft developed Kinect, a real-time RGB-D camera based on structured light, significantly promoted the development of 3D reconstruction. Han et al. [17] gave an overview of the computer vision and reconstruction method with Kinect.

Although RGB-D cameras offer plug-and-play convenience, their distance measurement capabilities and sensitivity to lighting conditions are not as robust as that of LiDAR. Hence, reconstruction solutions combining LiDAR and RGB cameras demonstrate advantages. The primary step in employing a LiDAR and camera integrated system for 3D reconstruction is the calibration of the setup. Classic methods of calibration include Bouguet’s camera calibration toolbox [18] and Zhang and Pless’s method based on a checkerboard [19]. Recently, with the advancement of ultra-wide-angle imaging sensors, numerous omnidirectional camera calibration techniques have also been developed. For instance, Scaramuzza et al. [20] utilized the association of hand-clicked points in a full 3D map with points in catadioptric images to achieve omnidirectional camera calibration. Miao et al. [21] proposed an effective targetless method to simultaneously calibrate the intrinsic parameters for the camera and the extrinsic parameters for the camera and LiDAR.

### 2.2. Geometry Reconstruction

The reconstruction of a static environment has been a subject of extensive research for an extended period. Modern methods predominantly rely on the signed distance function (SDF), which was first introduced by Curless and Levoy [22]. Subsequently, Rusinkiewicz presented the first real-time reconstruction method based on SDF [23]. Later, the introduction of low-cost RGB-D cameras by Microsoft brought the handheld-device-based reconstruction method KinectFusion into public view [24]. Afterwards, Whelan et al. proposed ElasticFusion with the truncated signed distance function (TSDF) [25], an enhanced version of SDF to achieve high-density reconstruction.

SDF-based methods operate at the voxel level. However, regular voxels may lack flexibility when dealing with large-scale environments that contain intricate details. Additionally, they are constrained by predefined volume sizes and resolutions. To address this issue, Fuhrmann and Goesele [26] proposed an adaptive octree data structure, known as the layered SDF, to support varying spatial resolutions. Zeng et al. [27] introduced a four-level hierarchy that stores the TSDF at the finest level. Steinbrucker et al. [28] presented a multiresolution data structure capable of real-time accumulation on a CPU.

To mitigate memory consumption, Nießner et al. [12] introduced the concept of voxel hashing. This innovation allows for theoretically infinite situations by organizing data into voxel blocks, whose addresses are indexed by a hash table. The primary challenge with hash tables is the issue of collisions. To address this, Kahler et al. [29] improved the hash table method to reduce the likelihood of collisions. Additionally, Prisacariu et al. [30] implemented the hash table framework with cross-platform support and achieved compatibility in different hardware and operating systems.

### 2.3. Texture Noise Correction

In the realm of texture noise, specular highlight is the most prevalent issue, drawing considerable research attention over an extended period. In recent years, many methods have been proposed to remove highlights from images. Predominantly, most of these methods are based on the dichromatic reflection model, which represents the image as a linear superposition of the specular reflection component and the diffuse reflection component. For instance, He et al. [1] used independent component analysis (ICA) to separate the specular and diffuse components from an image, while Yang et al. [2] employed a single image and a low-pass filter for the same purpose. Additionally, Shen et al. [3] approached highlight removal by computing the specular fractions of the image pixels with intensity ratio. Similarly, Fu et al. [4] focused on removing specular highlights by promoting sparsity in encoding coefficients and adhering to color mixing theories. Further, Yang et al. [5] separated reflection components by adjusting saturations of specular pixels to match diffuse-only pixels with the same diffuse chromaticity. Expanding upon these methods, Yamamoto et al. [6] used a nonlinear high-emphasis filter and a similarity function to improve the separation of reflection components, and Wei et al. [7] went a step further by not only separating highlights but also estimating the position of the light source, assuming that surface geometry is known. Guo et al. [31] tackled specular reflection by decomposing the transmitted and reflected layers for a sequence of images with strong structural priors. Recently, there has been a shift towards learning-based methods for removing specular highlights from images, such as the shadow/specular-aware (S-aware) network proposed by Jin et al. [8].

Another texture noise problem is the color inconsistency between frames caused by sensor pose changes, resulting in contouring phenomena on the model texture. To  address this issue, Li et al. [9] and Ye et al. [10] applied a local image translation on the image plane to diminish seams between frames. Chuang et al. [11] proposed a method of using Poisson equation in the gradient domain to hide seams and generate a convincing texture result.

While the aforementioned noise removal algorithms have made significant strides, there remain some constraints. Primarily, the majority of these techniques primarily focus on image processing and are rarely integrated directly into the 3D reconstruction workflow. Additionally, they fall short of tackling the aforementioned challenges (specular highlight, color inconsistency, and object occlusion) concurrently. Furthermore, these methods are tailored for specialized datasets, thereby limiting the broader applicability. In contrast, our method successfully addresses these challenges.

## 3. Methodology

Our entire process consists of three main stages. The first stage is the data organization phase, where we obtain a colored point cloud with luminance value through color space conversion, point cloud coloration, and multiframe registration. The second stage is the geometry reconstruction stage. Here, we employ the voxel hashing method [12] to build the geometry with the organized data obtained from the preceding stage. The final stage is the texture optimization stage, where we eliminate texture noise by fusing frames with the frame-voting rendering and the neighbor-aided rendering mechanisms. Our framework follows the pipeline illustrated in Figure 3.

### 3.1. Data Organization

To initiate the reconstruction process, we organize the input data into a dense colored point cloud with luminance value and mark all of the points with a source frame sequence number. This is achieved through color space conversion, point cloud coloration, and multiframe registration. In color space conversion, we convert RGB images into CIELAB images [32] to perform luminance comparison for subsequent texture rendering, as shown in Figure 4b. Within the CIELAB image, L channel values are used to represent the luminance of the frame. Compared with the three-channel RGB images, the utilization of the L channel from the CIELAB image allows for a more effective brightness comparison between frames. In point cloud coloration, we map LiDAR points to image pixels for point coloration using predetermined intrinsic and extrinsic matrices of the sensors. These matrices are accessed from an effective omnidirectional camera and using a non-repetitive LiDAR cocalibration method [21]. In frame registration, we employ the FAST-LIO [33] and generalized iterative closest point (GICP) algorithm [34] to determine the transformation matrices between frames to acquire a dense colored point cloud. In addition, the data utilized for the reconstruction must guarantee at least an 80% overlap between consecutive frames. This overlap ensures that each corner of the scene can be reconstructed using at least four to five frames, thereby providing a robust dataset for enhancing the quality of texture optimization in subsequent processing steps.

### 3.2. Geometry Reconstruction

During the process of geometry reconstruction, we apply voxel hashing [12], a memory-efficient method for large-scale scenes, to construct the geometry model. Voxel hashing is a two-level voxel data structure indexed by a hash table. A voxel block comprises 8×8×8 constant-sized voxels, which preserves the texture details of the model. The process of voxel hashing consists of two parts: hash table creation and point assignment.

In hash table creation, we calculate the voxel block coordinate for each LiDAR point and map the coordinate to the corresponding index using the hash function described in Formula (Equation 1). In the formula, *x*, *y*, and *z* are the coordinates of the voxel and μ is a predefined mask used to limit the range of hash values. Additionally, the values of the parameters p1, p2, and p3, which are large prime numbers, are determined through empirically driven settings to minimize collisions in the hash function. The output indexes form the hash table, serving as entries to the memory location of voxel blocks, as shown in Figure 5. Consequently, the hash table allows for the continuous storage of discrete voxel blocks, significantly improving memory efficiency.
(1)$$I\left(x,y,z\right)=\left(\left(x\cdot p_{1}\right)⊕\left(y\cdot p_{2}\right)⊕\left(z\cdot p_{3}\right)\right)\&\mu ;$$

The point assignment process begins with the allocation of memory space for point storage based on the number of points in each voxel block. Then, we assign the points to the corresponding voxel blocks according to the hash entries. Furthermore, we assign the the points to the voxel according to the relative position within the voxel block. The point information comprises not only point coordinate, color, and luminance values, but also the source frame number. The sequence number keeps track of the origin of each LiDAR point, enabling the discrimination of frames during the optimization rendering stage.

### 3.3. Texture Optimization

#### 3.3.1. Frame-Voting Rendering

The frame-voting rendering involves two steps: the calculation of the voxel target color and the color optimization. As mentioned in the previous data organization stage, the input data should ensure that there is a high overlap between frames covering the scene. Normally, the overall color and luminance value within a voxel are consistent, with only a minority of frames exhibiting significant color differences. Therefore, we can easily remove the color discrepancy by excluding outlier frames from the voxel. This method is particularly effective for addressing specular reflection-induced highlight texture noise. As the relative pose of the sensor and light source changes in different frames, the highlight location changes accordingly. This characteristic allows the frame-voting mechanism to effectively eliminate the highlight. The method also mitigates object occlusion. Moving entities, such as people or animals, and static object occlusion caused by occasional pose errors appear only in certain frames. Therefore, the majority of frames without occlusion can aid in filtering out these sporadic obstructions. Moreover, it resolves color inconsistency through point-by-point color optimization that ensures seamless color transitions between frames.

This paragraph presents the calculation process of the target color within a voxel. The procedure begins by filtering outliers at the frame level, followed by computing the average target color at the point level. Within this context, $L_{ij}$ represents the luminance value of the *j*th point in frame *i* within the voxel. We calculate the average luminance for each frame independently, using the formula $L_{i}=\frac{1}{n_{i}}\sum_{j=1}^{n_{i}}L_{ij}$, where $n_{i}$ is the number of points in frame *i*. Subsequently, the overall luminance of the voxel, denoted as $L_{mean}$, is calculated according to $L_{mean}=\frac{1}{n}\sum_{i=1}^{n}L_{i}$, and the variable range of luminance, denoted as $L_{var}$, is calculated according to $L_{var}=\sqrt{\frac{1}{n}\sum_{i=1}^{n}{\left(L_{i}-L_{mean}\right)}^{2}}$, where *n* is the number of frames within this voxel.
(2)$$C_{target}=\frac{1}{N}\sum_{L_{i}-L_{mean}\leq L_{var}}\sum_{j=1}^{n_{i}}C_{ij},R,G,B\in C$$ In Formula (Equation 2), we identify frames that meet the condition $L_{i}-L_{mean}\leq L_{var}$ to calculate the target color for the voxel. Here, *N* refers to the total number of points contained in the frames that satisfy the selection criterion. Based on this calculation method, frames with a larger number of points have a greater influence on the final result of the target color.

During the color optimization process, we optimize the point colors based on the previously calculated target color. Specifically, only the points whose brightness value exceeds the variance range, expressed as $L_{ij}-L_{mean}>L_{var}$, need to be updated to the target color. This selective optimization preserves the original texture details, ensuring a more realistic and visually pleasing rendering outcome.

#### 3.3.2. Neighbor-Aided Rendering

Indeed, the frame-voting rendering effectively mitigates texture noise in many scenarios. However, certain voxels lack a sufficient number of points for self-rendering due to the random distribution of points within the real scene, as depicted in Figure 6. To address this limitation, we introduce the neighbor-aided rendering mechanism for target color calculation. As the name implies, it leverages neighboring voxels to provide luminance and color information for the central voxel which has insufficient points. This method ensures a more complete and precise rendering outcome for the entire scene.

The main idea of the neighbor-aided rendering is to cluster neighboring voxels based on their overall luminance, and then select the cluster with the largest number of points to calculate the ultimate target color. This method allows us to accurately perform color compensation for the central voxel in scenes where neighboring voxels have significant color differences, such as at the boundaries between a white wall and a door where adjacent voxels display varying colors. The operational steps of the approach are demonstrated in Algorithm 1. Initially, we start with no groups, and both the group number gn and the voxel index *i* are initialized to 0. Moving forward, for each neighboring voxel $v_{i}$, the algorithm checks existing groups to determine if any group exhibits a luminance level close to that of $v_{i}$. If such a group exists, the algorithm adds $v_{i}$ to that group and updates the overall luminance of the group; if not, a new group $G_{gn}$ is created, $v_{i}$ is included within it, and the group number gn is incremented. Upon completing the clustering process for all neighboring voxels, the group $G_{optimal}$ with the highest point count is selected. Finally, the algorithm calculates the ultimate target color $C_{target}$ for the voxel utilizing the average color of the points contained within $G_{optimal}$.
**Algorithm 1:** Neighbor-Aided Rendering
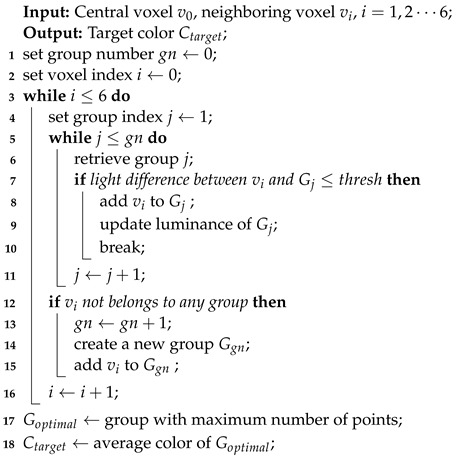


When the central voxel locates in the outermost layer of the voxel block, part of the neighboring voxels are situated in adjacent voxel blocks, as depicted in Figure 7. However, in computer memory, the physically adjacent voxel blocks are stored at distant memory addresses which are challenging to access directly. To overcome this issue, we incorporate the hash entries of adjacent voxel blocks as a component of the information stored within the data structure of each voxel block. This design facilitates efficient retrieval of neighboring information during the rendering process.

## 4. Experiment

In this section, we showcase the experimental results along with an efficiency analysis and visual representations of rendering optimization outcomes. In the efficiency analysis, we begin by introducing the sensor setup and describing the data characteristics utilized in the experiment. Subsequently, we conduct an analysis of both memory and time efficiency within the reconstruction process. Regarding rendering optimization representation, we illustrate the effects of employing frame-voting rendering and neighbor-aided rendering mechanisms to mitigate texture noise, supported by experimental results.

### 4.1. Experimental Environment, Equipment, and Data

During the experiment, we employ an omnidirectional camera and the Mid-360 LiDAR (Livox Tech Co., Ltd., Shenzhen, China) for data collection, allowing us to capture a 360-degree field of view and ensure high overlap between frames, as depicted in Figure 8. The whole tasks are evaluated on an Intel i7-10700K CPU @ 3.80 GHz with 16 GB memory.

Figure 9 visually represents the organization of our data, highlighting the collection process across four distinct spots. Each spot consists of data captured from five specified poses, determined by our gimbal setup. For a clearer understanding of the spatial relationships, we explain that the transformation matrices between poses are derived from the gimbal’s configuration. Additionally, the matrices between different spots were initially generated using the Fast-LIO [33] algorithm and further refined using the GICP [34] method. This detailed representation aims to provide a comprehensive understanding of our data collection and processing methodology.

### 4.2. Efficiency Analysis

Table 1 presents detailed information about our data, the primary parameter settings for voxel hashing, and the associated voxel block costs. Notably, we set the voxel resolution to 0.05 m deliberately. A lower resolution would negatively affect appearance continuity, whereas a higher one would result in insufficient points for effective voxel-based texture optimization. Importantly, by implementing the voxel hashing data structure, we significantly reduced the number of voxel blocks from 44,000 to 11,284, compared with full scene coverage without hash mapping. This amounts to an impressive 75% reduction in computer memory consumption.

Table 2 summarizes the primary stages of the reconstruction process along with their corresponding time efficiencies. The creation of the hash table is accomplished in about 1.35 s. Additionally, the allocation of memory space for voxel blocks and the assignment of points to the relevant voxel blocks are carried out in approximately 5.93 s. The computation for frame-voting rendering takes around 21.60 s, while the neighbor-aided rendering procedure consumes roughly 21.37 s. The processing of the entire dataset is completed within 60 s, dealing with nearly 70 million points. This equates to a processing rate of over 1 million points per second, suggesting that the solution has the capability for real-time processing.

### 4.3. Experiment Results of Texture Optimization

#### 4.3.1. Results on Frame-Voting Rendering

In this subsection, we present the outcomes of our rendering optimization method. Figure 10 illustrates the results of highlight noise correction in a scene where prominent light spots and halation are caused by the specular reflection of the camera lens. The comparison between the original point cloud and the optimized point cloud is displayed on the left side, while the zoomed-in sections of the right side provide a clearer view of the highlight noise correction effect. We can see that the highlight noise and halation phenomena were significantly mitigated, and the quality of the texture was effectively improved.

Figure 11 illustrates the efficacy of our approach in mitigating texture noise arising from object occlusion. The upper-right image presents the original RGB image of the scene, highlighting the area with chair and table occlusion. On the left side, a comparison is made between the original point cloud and the optimized point cloud, while the lower-right figures provide a closer examination of the contrasting results. Through the frame-voting rendering mechanism, object occlusion caused by occasional pose errors is well filtered out.

#### 4.3.2. Results on Neighbor-Aided Rendering

Figure 12 is the experimental result of the neighbor-aided rendering mechanism, which is shown in the following order: the original image, the result without neighbor-aided rendering, and the result with neighbor-aided optimization. From (a) to (b), the image quality is obviously enhanced by applying the frame-voting rendering. From (b) to (c), the result demonstrates that the neighbor-aided rendering significantly reduces the texture noise that cannot be removed directly due to the insufficient number of points inside the voxel.

#### 4.3.3. Comparing Results of Highlight Removal

To demonstrate the effectiveness of our method in eliminating highlights, we conducted comparisons with other highlight removal techniques. Since the majority of these methods are designed for image processing rather than 3D LiDAR cloud data, we projected the reconstructed model onto images. The results are presented in Figure 13, where (a) represents the projection of the reconstructed model without texture optimization; (b) represents the projection of our method, and it is after texture optimization; and (c), (d), (e), and (f) depict the results of highlight removal modifications applied to the projection of a raw model using the techniques of Yang et al. (2010) [2], Shen et al. (2013) [3], Fu et al. (2019) [4], and Jin et al. (2023) [8], respectively. Our approach effectively eliminates texture noise while preserving the overall image brightness, contrast, saturation, and structural information, thus preventing significant alterations that could lead to image distortion. To gauge image quality in highlight removal tasks, we utilize the SSIM (structure similarity index), PSNR (peak signal-to-noise ratio) [35], and FSIM (feature similarity index) [36] metrics, with the corresponding numerical results provided in Table 3.

## 5. Conclusions

In this paper, we proposed an omnidirectional-sensor-system-based texture noise correction framework for large-scale 3D reconstruction according to data organization, geometry reconstruction, and texture optimization. It fuses LiDAR points and RGB images into a colored point cloud with luminance value and constructs a space-efficient geometry model using voxel hashing [12]. Most importantly, it simultaneously reduces multivariate mixed noise such as highlight problems, frame color inconsistency, and object occlusion through frame-voting and neighbor-aided mechanisms.

Our approach holds significant promise in the surveying and mapping domain, as it efficiently handles large volumes of input data and improves texture quality degraded by sensor performance limitations and environmental disturbance. More specifically, our research offers substantial advancements in geographic information systems (GIS) development and cultural heritage preservation, effectively addressing challenges such as occlusion in urban environments and illuminance limitations in indoor heritage sites. The system currently places primary emphasis on texture optimization and requires the precision of transformation matrices between frames. Therefore, in the future, we will focus more on the accuracy of geometry reconstruction while ensuring texture optimization. Considering our texture optimization’s capability to process up to one million points per second, it holds significant potential for real-time processing. Current SLAM frameworks that utilize image and LiDAR data for real-time applications, such as FAST-LIO [33] and R3LIVE [37], excel in geometric accuracy but often do not focus as much on texture quality. By integrating the strengths of our texture optimization approach with the precise geometric localization and mapping capabilities of these SLAM frameworks, we envision creating a more comprehensive and enhanced reconstruction framework in the future.

## Figures and Tables

**Figure 1 sensors-24-00078-f001:**
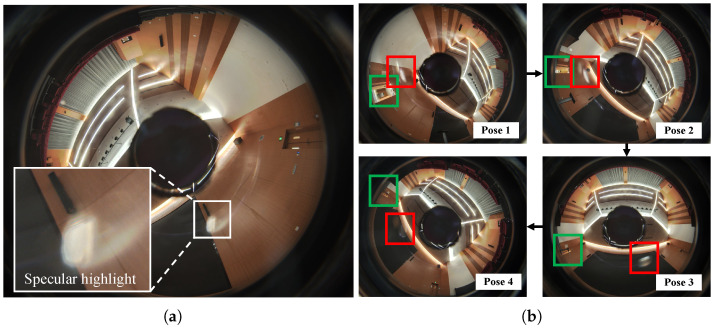
(**a**) Specular highlight phenomenon. (**b**) The position of the highlight areas in the image changes with the variation of the sensor pose. In the image, the red box indicates the most prominent highlight noise, and the green box indicates the door, which serves as a positional reference.

**Figure 2 sensors-24-00078-f002:**
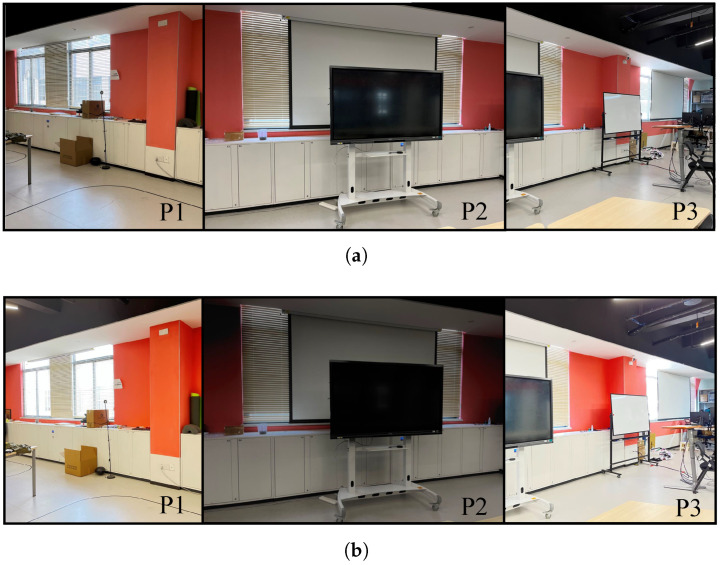
Color inconsistency phenomenon. P1–P3 are three consecutive images in terms of position. (**a**) Normal situation with consistent color between frames. (**b**) Inconsistent color between frames caused by variations in the intensity of the light source or changes in its relative position to the sensor.

**Figure 3 sensors-24-00078-f003:**
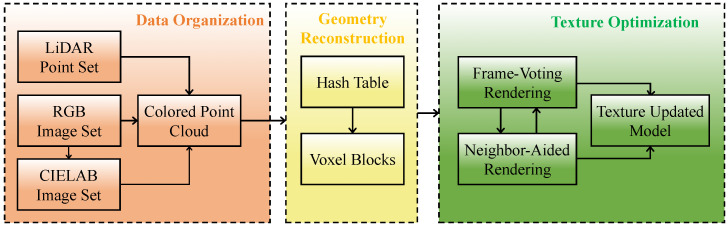
Pipeline of the whole process, consisting of data organization, geometry reconstruction, and texture optimization.

**Figure 4 sensors-24-00078-f004:**
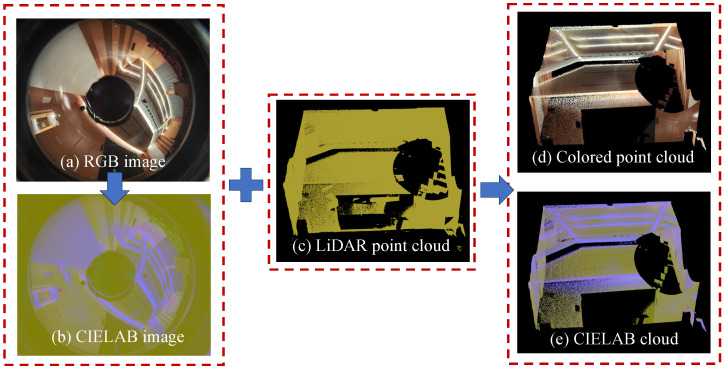
Process flow of data organization. (**a**) RGB image. (**b**) CIELAB color space image transformed from RGB image, which facilitates luminance evaluation in the subsequent section of our work. (**c**) LiDAR point cloud. (**d**) Fusion of LiDAR point cloud with RGB image. (**e**) Fusion of LiDAR point cloud with CIELAB color space image.

**Figure 5 sensors-24-00078-f005:**
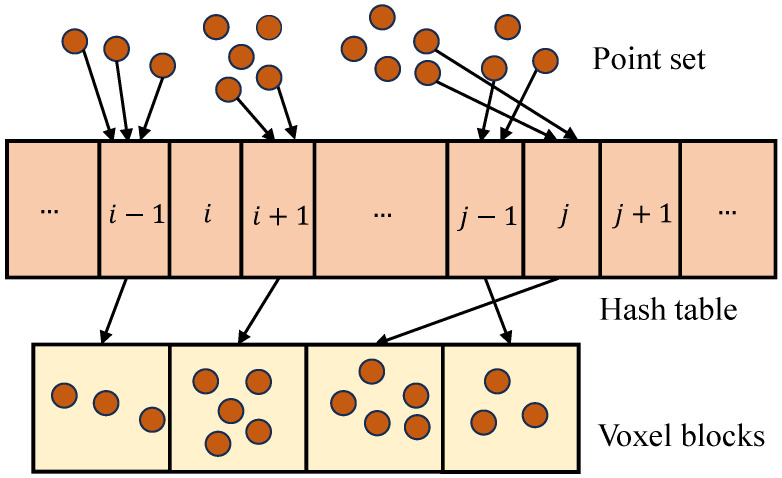
Voxel hashing schematic. The mapping between point coordinates and voxel block indices is achieved through a hash table, thereby efficiently allocating points while making reasonable use of computer storage resources.

**Figure 6 sensors-24-00078-f006:**
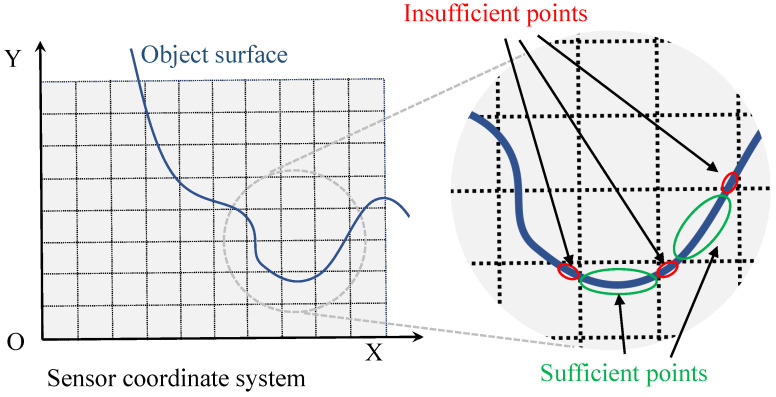
Motivation for proposing neighbor-aided rendering mechanism: points are randomly distributed in voxels; thus, some voxels lack insufficient points for self-optimization.

**Figure 7 sensors-24-00078-f007:**
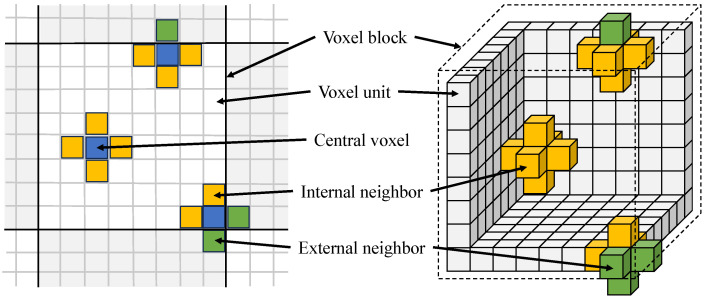
Neighbor-aided rendering mechanism. The figure illustrates the configuration of a voxel block and the interconnections between adjacent voxels.

**Figure 8 sensors-24-00078-f008:**
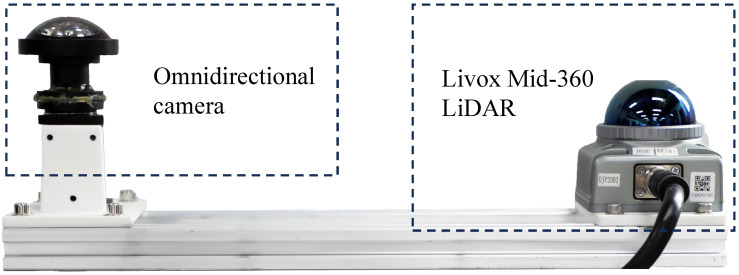
Sensor setup for data collection.

**Figure 9 sensors-24-00078-f009:**
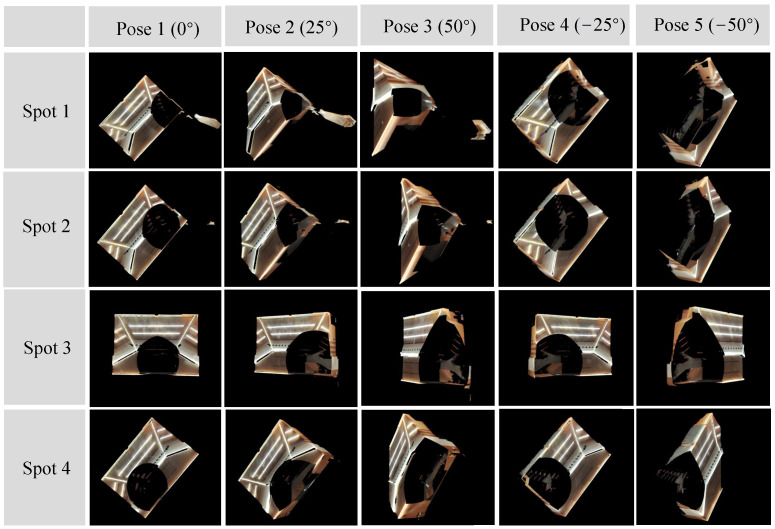
Input data. The dataset consists of four spots, and each spot consists of five specified poses.

**Figure 10 sensors-24-00078-f010:**
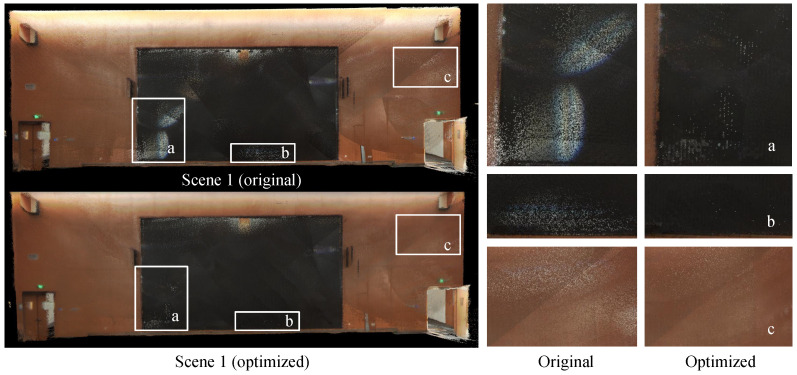
Highlight noise correction in scene 1 according to frame-voting rendering. Regions (**a**)–(**c**) present specular highlights phenomenon on the screen and wall surfaces in the scene.

**Figure 11 sensors-24-00078-f011:**
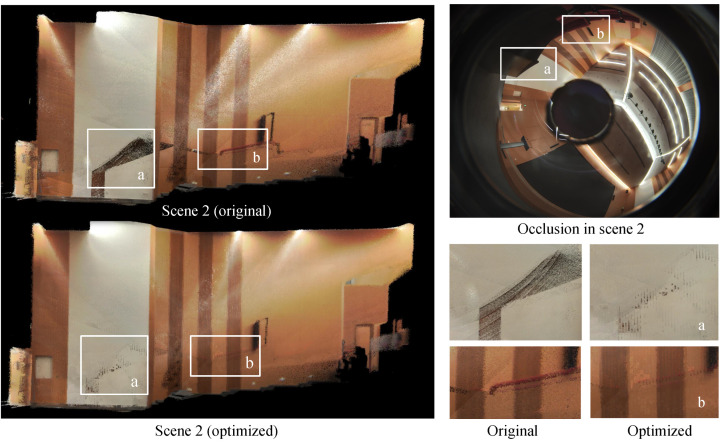
Elimination of object occlusion in scene 2 with frame-voting rendering. (**a**) Comparison diagram of the elimination of misimaging caused by table occlusion. (**b**) Comparison diagram of the elimination of misimaging caused by chair occlusion.

**Figure 12 sensors-24-00078-f012:**
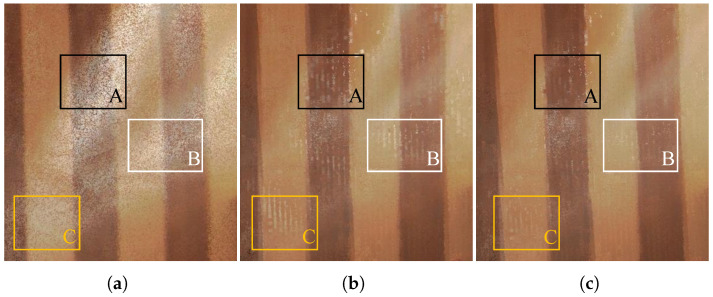
Enhanced outcome with neighbor-aided optimization. Regions A–C exhibite pronounced contrastive effects. (**a**) Demonstration area of the original point cloud containing numerous types of texture noise. (**b**) The result optimized using only frame-voting rendering. (**c**) The result optimized further with neighbor-aided rendering.

**Figure 13 sensors-24-00078-f013:**
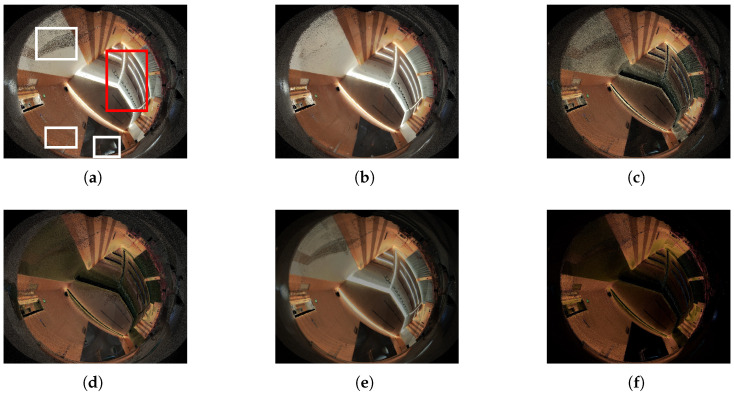
Comparing results of highlight removal method. (**a**) Projection of raw model (input). The white boxes indicate areas with noise that should be corrected. The red box indicates area that should not be corrected (lights). (**b**) Projection of texture optimized model (ours). (**c**) Yang et al. (2010) [2]. (**d**) Shen et al. (2013) [3]. (**e**) Fu et al. (2019) [4]. (**f**) Jin et al. (2023) [8].

**Table 1 sensors-24-00078-t001:** Analysis of data characteristics and memory efficiency.

Data	Scale
Frame number	20
Point number	69,740,000
Scene size	22m×16m×8m
Voxel resolution	0.05 m
Voxel block size	8×8×8
VB number without hash mapping	44,000
VB number with hash mapping	11,284

**Table 2 sensors-24-00078-t002:** Time efficiency analysis for key stages.

Stage	Computation Time(s)
Hash table creation	1.35
Point assignment	5.93
Frame-voting rendering	21.60
Neighbor-aided rendering	21.37

**Table 3 sensors-24-00078-t003:** Image quality evaluation on highlight removal.

Methods	SSIM ↑	PSNR (dB) ↑	FSIM ↑
Yang et al. (2010) [2]	0.6451	13.7009	0.8373
Shen et al. (2013) [3]	0.6091	12.0770	0.8134
Fu et al. (2019) [4]	0.7907	16.5348	0.9086
Jin et al. (2023) [8]	0.2492	9.8382	0.7187
Ours	**0.8852**	**21.6157**	**0.9153**

## Data Availability

The data presented in this study are available from the corresponding author upon reasonable request.

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
