# Peer review of "Omnidirectional-Sensor-System-Based Texture Noise Correction in Large-Scale 3D Reconstruction"

_sensors, 2023, doi:10.3390/s24010078_

Round 1
Reviewer 1 Report
Comments and Suggestions for Authors
This paper developed an interesting omnidirectional sensor system based texture noise correction in large-scale 3D reconstruction. Three key procedures-data organization, geometry reconstruction and texture mapping have been introduced carefully and thoroughly.
The article is well-organized and the main idea has been conveyed clearly, which is of great significance in technical applications. I have one major problems for the authors: as we know, The major error sources of Lidar/ToF cameras come from multipath interference (MPI) and reflectivity related distance variations., as pointed out in the work of “Accurate Depth Recovery Method Based on the Fusion of Time-of-Flight and Dot-Coded Structured Light”. Here the authors have solved the specular light problems, how about the MPI problem? I think it would be more meaningful if the authors consider and discuss this in their work.
This paper developed an interesting omnidirectional sensor system based texture noise correction in large-scale 3D reconstruction. Three key procedures-data organization, geometry reconstruction and texture mapping have been introduced carefully and thoroughly.
The article is well-organized and the main idea has been conveyed clearly, which is of great significance in technical applications. I have one major problems for the authors: as we know, The major error sources of Lidar/ToF cameras come from multipath interference (MPI) and reflectivity related distance variations., as pointed out in the work of “Accurate Depth Recovery Method Based on the Fusion of Time-of-Flight and Dot-Coded Structured Light”. Here the authors have solved the specular light problems, how about the MPI problem? I think it would be more meaningful if the authors consider and discuss this in their work.

Minor editing of English language required
Author Response
Thank you for your valuable advice. We carefully reviewed the paper "Accurate Depth Recovery Method Based on the Fusion of Time-of-Flight and Dot-Coded Structured Light" that you recommended. The fusion of TOF and DCSL technologies indeed provided us with valuable insights.
We acknowledge that multipath interference (MPI) can significantly impact measured distances. In our experiments, our setup consists of a LiDAR for depth information and a camera for color information. This combination inevitably introduces the challenge of MPI-induced depth errors. However, it's worth noting that we utilize a non-repetitive LiDAR, developed by LIVOX, which helps mitigate MPI-related issues to a certain extent.
Non-repetitive LiDAR offers advantages due to its ability to achieve denser scene coverage. This means that even if a specific point is affected by MPI, the surrounding points can contribute additional depth information, partially compensating for the disturbance. While its effectiveness may be reduced in scenes with large reflective surfaces, non-repetitive LiDAR still outperforms traditional repetitive scanning LiDAR.
Looking ahead, we are considering the integration of image data into our geometric reconstruction process. This integration will serve to complement and refine the geometric information obtained from LiDAR, helping us address geometric errors more effectively caused by MPI.
We hope that our revisions adequately address your concerns. Thank you again for your constructive feedback, which has helped strengthen our paper.
Reviewer 2 Report
Comments and Suggestions for Authors
In this paper, the authors proposed an omnidirectional sensor system based texture noise correction framework for large-scale 3D reconstruction according to data organization, geometry reconstruction, and texture optimization. The approach looks fine, but the experimental demonstration is not sufficient. There are quite a few points to be figured out.
1. Page 6, in Figure 4, it seems that there are two different sets of scene images. If this is only one set of images, explain the position relationship between the two sets of images. If this is not a set of images, it should display the processing of the two sets of images separately to make the image conversion clearer.
2. In Section 3.2, the hash functions used should be described in detail.
3. Page 8, The final target color is determined by directly averaging the color of the points in the group after clustering. If weighted function is adopted, will the effect be better?
4. Is the proposed texture noise correction framework universal? Why?
5. In Section 3.3.1, how to resolve the effect of noise points on the neighbor-aided rendering?
6. In the experiments, the results of triangular mesh 3D models obtained by different methods should be clearly displayed.
Comments on the Quality of English LanguageThe authors should improve their language in the manuscript, and English spell check is required.
Author Response
Thank you for your valuable feedback and suggestions. We appreciate the opportunity to improve our manuscript based on your insightful comments.
1.Modification of Figure 4.
We sincerely apologize for any confusion caused by the representation in Figure 4. To address this issue and provide greater clarity, we have made adjustments to the figure.
Figure 4 is intended to depict a single dataset, comprising (a)RGB images and (c)LiDAR point clouds, which are frame-to-frame corresponding. It is important to note that LiDAR point clouds are responsible for providing depth information, while RGB images contribute color information. Consequently, subfigure (d) illustrates the fusion of LiDAR point cloud with RGB image.
Additionally, subfigure (b) depicts the CIELAB color space image tranformed from RGB image, which facilitates luminance evaluation in the subsequent section of our work. Finally, subfigure (e) represents the fusion of LiDAR point cloud with CIELAB color space image. We hope these clarifications help in better understanding the content and purpose of Figure 4.
2.Description of hash function.
We have enhanced Section 3.2 by incorporating a detailed formula and additional explanations about the hash function. This section now elucidates how voxel block coordinates serve as input data, and how the hash function translates these into indices corresponding to physical memory addresses. Furthermore, we have clarified that the parameters p1, p2, p3 are large prime numbers, selected based on empirical evidence to effectively minimize collisions within the hash function.
3.If weighted function is adopted, will the effect of target color be better?
Your comment regarding the use of weight is very insightful and is what we attach great importance to. Actually, we have carefully considered the weight allocation for each frame.
In our technique, we first filter outliers at the frame level, and then compute the average target color at the point level. As we all know, frames with a higher point count naturally exert a more substantial influence compared to frames with fewer points. The averaging of all the remained points can be regarded as a method of assigning a 'point count' weight to each frame.
We hope this response has adequately addressed your question. We are open to any further clarification or discussion regarding this matter.
4.Is the proposed texture noise correction framework universal? Why?
Thank you for your question regarding the universality of our method. We developed our approach based on theoretical principles and design considerations that are broadly applicable in the field. We recognize that our current method validation, being based on a single dataset, may not fully demonstrate its universality. However, the promising results we have obtained so far suggest potential applicability across a wider range of scenarios. In future work, we plan to adapt our method to a real-time framework, which focuses more on accurate geometry reconstruction. This will enable us to verify the effectiveness of our method in texture optimization with real-time input data, thereby providing a more comprehensive assessment of its universality.
5.In Section 3.3.1, how to resolve the effect of noise points on the neighbor-aided rendering?
If you are referring to the scenario where the neighboring voxel itself contains noise points, we believe this should not pose a significant issue. When a voxel contains a sufficient number of points, it has the capability to mitigate occasional noise points through the target color calculation as outlined in Section 3.3.1.
If, on the other hand, you are concerned about a substantial color disparity between neighboring voxels, such as the scenario with a white wall adjacent to a brown door, our algorithm detailed in Section 3.3.2 is specifically designed to handle such cases. Instead of averaging all neighboring voxel colors, our approach categorizes neighboring voxels based on color similarity. The color of the central voxel is then determined based on the color that has the highest weight in terms of point count. For instance, if some neighboring voxels tend toward white while others lean toward brown, and the total number of points in the white group is larger, the final decision for the central voxel's color will be white.
6.In the experiments, the results of triangular mesh 3D models obtained by different methods should be clearly displayed.
Thank you for your suggestion to present triangular mesh models in the experiment. We appreciate the advantages of mesh representation, particularly in terms of storage efficiency. However, there are some limitations in the current meshing results of our method, primarily due to the inherent thickness errors (2-3cm) in point cloud data collected in large-scale environments. These errors can lead to the generation of meshes with less smooth appearances.
It's important to note that some prevalent large-scale SLAM methods, which excel in geometric reconstruction, primarily operate using point cloud representations. Examples include FAST-LIO and R3LIVE. Therefore, eliminating the geometric error is the first step before converting lidar point cloud to meshes in large-scale environment. Typically, LiDAR data quality can be optimized through point cloud filtering, or by fusing data with other types of sensors, such as Inertial Measurement Units (IMU), GPS, etc., to enhance the overall system's accuracy and reliability. This will be the focus of our future research direction. Because we also hope that point cloud data collected in large-scale scenes can generate high-quality meshes, facilitating more convenient and widespread application of 3D reconstruction technology.
We hope that our revisions adequately address your concerns. Thank you again for your constructive feedback, which has helped strengthen our paper.
Reviewer 3 Report
Comments and Suggestions for Authors
The article focuses on developing a texture noise correction system in three-dimensional reconstruction using omnidirectional sensors. Contemporary challenges associated with sensor limitations and environmental disturbances drive the search for innovative solutions in this field. The authors present a comprehensive approach applicable to large spatial scales. The strength of the article lies in the proposed comprehensive 3D reconstruction system that focuses on eliminating texture noise. Particularly innovative are the frame-voting rendering and neighbor-aided rendering mechanisms, effectively removing various disturbances such as specular highlights and color inconsistency. The innovations presented by the authors include three key aspects. Firstly, a comprehensive 3D reconstruction system based on omnidirectional sensors. Secondly, the frame-voting rendering mechanism eliminates various texture noises. Thirdly, the neighbor-aided rendering mechanism optimizes color in areas with an insufficient number of points for self-texture optimization.
The Introduction section requires additions. It should include descriptions or references to existing solutions related to noise correction systems. The literature review, both in scientific and industry contexts, also needs to be supplemented and reinforced. This has translated into a lack of discussion and a missing critical analysis of the obtained results, including a comparison with other solutions.
The article is clear and well-structured. However, a more detailed presentation of experimental results and their interpretation in the context of practical system application would be beneficial. The lack of specific information about real-world use cases and an in-depth analysis of potential practical challenges may be considered a limitation. Additionally, expanding the discussion on the limitations of the proposed system could enhance the article. Furthermore, expanding the section on future research directions could be valuable.
Author Response
Thank you for your valuable feedback and suggestions. We appreciate the opportunity to improve our manuscript based on your insightful comments.
1.Introduction: adding descriptions or references to existing solutions related to noise correction systems.
Thank you for your insightful comment regarding the inclusion of descriptions and references to existing noise correction systems. Following your suggestion, we have added a chart in the Introduction section that provides an overview of existing solutions in this domain. This addition not only outlines the key methods currently in use but also briefly discusses their limitations. This serves as a segue into the introduction of our framework, highlighting how our approach addresses these identified gaps.
2.Literature review supplementation: both in scientific and industry contexts.
To thoroughly address your concerns, we have expanded the Literature Review section to include a comprehensive array of highlight removal methods, ensuring a more exhaustive discussion of the topic.
3.Experiment and results: a lack of discussion and a missing critical analysis of the obtained results, including a comparison with other solutions.
We have addressed this concern by incorporating a dedicated section 4.3.3 for highlight removal comparison. Within this section, we have presented the comparison outcomes between various highlight removal methods, accompanied by a comprehensive evaluation based on established metrics such as SSIM, PSNR, and FSIM. This evaluation effectively underscores our method's proficiency in preserving overall image contrast and structural integrity.
4.Conclusion.
According to your valuable reminder, we have now included the following discussion in the Conclusion section of the article to enhance its comprehensiveness.
(1)Practical system applications and real-world use cases.
Our research on 3D reconstruction of large-scale environments has significant practical implications, particularly in fields such as Geographic Information Systems (GIS) and cultural heritage preservation. In GIS developing, there are challenges like occlusions caused by pedestrians and vehicles in the data collection stage. Our multi-frame fusion techniques are specifically designed to tackle these occlusions effectively.
Additionally, our work is highly relevant in the context of cultural heritage documentation and preservation, where illuminance limitations are a major concern. For instance, in indoor heritage sites like palaces or tombs, the absence of natural light sources makes lighting conditions a primary challenge. The use of artificial lighting to counter this issue often results in inconsistent lighting and high-intensity noise in the reconstructed images. Our research directly addresses these challenges, providing solutions that enhance the quality of indoor heritage site reconstructions, thus contributing valuable tools for preserving our cultural heritage.
(2)Potential practical challenges.
In practical applications, 3D reconstruction faces several potential challenges. For applications generating large volumes of data, like GIS, storing highly detailed environmental information within constrained storage resources can be a significant challenge. Additionally, when it comes to heritage reconstruction, more sophisticated sensor-carrying devices such as unmanned aerial vehicles may be required for data collection, leading to increased complexity in geometric calculation.
(3)Limitations of the proposed system.
The system currently places a primary emphasis on texture optimization and requires the precision of transformation matrices between frames.
Future research will focus more on the accuracy of geometric reconstruction while ensuring texture optimization.
(4)Future research directions.
Considering our texture optimization's capability to process up to one million points per second, it holds significant potential for real-time processing. Current SLAM frameworks that utilize image and LiDAR data for real-time applications, such as FAST-LIO and R3LIVE, excel in geometric accuracy but often do not focus as much on texture quality. By integrating the strengths of our texture optimization approach with the precise geometric localization and mapping capabilities of these SLAM frameworks, we envision creating a more comprehensive and enhanced reconstruction framework in the future.
We hope that our revisions adequately address your concerns. Thank you again for your constructive feedback, which has helped strengthen our paper.
Reviewer 4 Report
Comments and Suggestions for Authors
The article deals with the problem of 3D reconstruction of LiDAR data. The topic is topical as the use of LiDAR technology is expanding into many scientific disciplines and practices. The objectives and workflow of the paper are clearly defined.
Comments:
What types of data were collected in the experimental part?
Were different surfaces of objects tested and also the coloration of the surface of the objects?
How did the distance of the objects affect the accuracy of the resulting model?
I recommend to describe the experimental part in more detail, especially in terms of input data.
Did you also use statistical methods to evaluate the method? If so, these should be described.
Author Response
Thank you for your valuable feedback and suggestions. We appreciate the opportunity to improve our manuscript based on your insightful comments.
1.Data type.
We primarily gathered two types of data: LiDAR and RGB images. The LiDAR data provided us with precise distance measurements and the RGB images were used to capture the color and texture details of the environment.
2.Were different surfaces of objects tested and also the coloration of the surface of the objects?
Thank you for your insightful question regarding the types of surfaces in the experiments. Our approach, grounded in broad theoretical principles and design considerations, was indeed applied to a variety of surfaces in our experimental setup. This includes surfaces such as walls, wooden doors, projection screens and fabric chairs. The majority of these surfaces demonstrated promising results in terms of texture quality. Notably, projection screens, which are typically prone to highlight noise, were effectively processed using our method, as evidenced in Figure 10. We believe these diverse applications highlight the adaptability and effectiveness of our approach in handling a range of surface types and conditions.
However, we acknowledge that our current method validation, being based on a single dataset, may not fully demonstrate its universality. To address this, we plan to adapt our approach to a real-time framework in future work. This adaptation will allow us to test our method's effectiveness in texture optimization with real-time input data, providing a more comprehensive assessment of its universality and applicability.
3.How did the distance of the objects affect the accuracy of the resulting model?
Your question regarding the impact of object distance on the accuracy of the model is indeed important. It is a well-established fact in our field that increased distances from the objects can lead to larger geometric errors. This is particularly true for large-scale scenes, where meticulous data preprocessing is crucial to ensure the precise generation of transformation matrices, thus maintaining the accuracy of the model.
Regarding the system's ability to capture texture colors, our findings indicate that the influence of object distance is relatively minimal. It does not significantly impact the overall accuracy of the model, as the texture color capture remains consistent even at varying distances.
4.More detailed description of the experiment (especially data).
In response to your request for enhanced clarity regarding our experimental data, we have added a new figure to Section 4.1. This figure visually represents the organization of our data, highlighting the collection process across four distinct spots. Each spot consists of data captured from five specified poses, determined by our gimbal setup.
For a clearer understanding of the spatial relationships, we explain that the transformation matrices between poses are derived from the gimbal's configuration. Additionally, the matrices between different spots were initially generated using the Fast-LIO algorithm and further refined using the GICP (Generalized Iterative Closest Point) method. This detailed representation aims to provide a comprehensive understanding of our data collection and processing methodology.
We trust that this additional visualization and information will enhance the clarity and understanding of our experimental data.
5.Statistical evaluation.
Thank you for your suggestion. We have addressed this concern by incorporating a dedicated section 4.3.3 for highlight removal comparison. Within this section, we have presented the comparison outcomes between various highlight removal methods, accompanied by a comprehensive evaluation based on established metrics such as SSIM, PSNR, and FSIM. This evaluation effectively underscores our method's proficiency in preserving overall image contrast and structural integrity.
We hope that our revisions adequately address your concerns. Thank you again for your constructive feedback, which has helped strengthen our paper.
Round 2
Reviewer 2 Report
Comments and Suggestions for Authors
The authors have corrected some algorithm derivation errors and expression errors. I agree to accept this manuscript.
Comments on the Quality of English LanguageThe author has carefully corrected errors in the language, and the quality of English language is better than before.
Reviewer 4 Report
Comments and Suggestions for Authors
I thank the authors for drafting the answers, answering all my questions, and completing the manuscript.